# Aspiring Antifungals: Review of Current Antifungal Pipeline Developments

**DOI:** 10.3390/jof6010028

**Published:** 2020-02-25

**Authors:** Thomas J. Gintjee, Monica A. Donnelley, George R. Thompson

**Affiliations:** 1Department of Inpatient Pharmacy, University of California – Davis, 2315 Stockton Blvd., Room 1310, Sacramento, CA 95817, USA; tjgintjee@ucdavis.edu (T.J.G.); mdonnelley@ucdavis.edu (M.A.D.); 2College of Pharmacy, Touro University, 1310 Club Drive, Vallejo, CA 94592, USA; 3Department of Medicine, Division of Infectious Diseases and Department of Medical Microbiology and Immunology, University of California – Davis, 4150 V Street; Suite G500, Sacramento, CA 95817, USA

**Keywords:** invasive fungal infections, antifungal drugs, pharmacodynamics, novel therapies, review

## Abstract

Invasive fungal infections are associated with significant morbidity and mortality, and their management is restricted to a variety of agents from five established classes of antifungal medication. In practice, existing antifungal agents are often constrained by dose-limiting toxicities, drug interactions, and the routes of administration. An increasing prevalence of invasive fungal infections along with rising rates of resistance and the practical limitations of existing agents has created a demand for the development of new antifungals, particularly those with novel mechanisms of action. This article reviews antifungal agents currently in various stages of clinical development. New additions to existing antifungal classes will be discussed, including SUBA-itraconazole, a highly bioavailable azole, and amphotericin B cochleate, an oral amphotericin formulation, as well as rezafungin, a long-acting echinocandin capable of once-weekly administration. Additionally, novel first-in-class agents such as ibrexafungerp, an oral glucan synthase inhibitor with activity against various resistant fungal isolates, and olorofim, a pyrimidine synthesis inhibitor with a broad spectrum of activity and oral formulation, will be reviewed. Various other innovative antifungal agents and classes, including MGCD290, tetrazoles, and fosmanogepix, will also be examined.

## 1. Introduction

There are currently five classes of antifungal agents used in the treatment of systemic mycoses: polyenes (amphotericin B), azoles (fluconazole, itraconazole, posaconazole, voriconazole, and isavuconazole), echinocandins (caspofungin, micafungin and anidulafungin), allylamines (terbinafine), and antimetabolites (flucytosine).

Polyenes achieve fungicidal activity by binding ergosterol in the cell membrane, resulting in increased permeability and the leakage of intracellular components, which subsequently leads to cell death (Figure 1) [1,2]. Amphotericin B is the most clinically relevant polyene for invasive fungal infections, and maintains a broad spectrum of fungicidal activity, covering yeasts, moulds, and dimorphic fungi. Its use in practice is limited by a lack of an oral formulation, infusion reactions, and significant dose-limiting toxicities such as nephrotoxicity. The development of several lipid-based formulations and co-administration with normal saline has improved patient tolerability but has not completely eliminated toxicities. Despite these drawbacks, amphotericin B sees consistent clinical use as empiric coverage of invasive fungal infections until a more tolerable therapy or formulation can be identified.

Similar to polyenes, azoles target ergosterol to achieve fungicidal activity [2,3]. They specifically inhibit ergosterol synthesis by inhibiting the lanosterol 14α-demethylase enzyme (Figure 1). Potent fungicidal activity and a broad spectrum of coverage as a class have made azoles first-line therapy for the treatment and/or prophylaxis of many invasive fungal infections. Many azoles are conveniently available as intravenous and oral formulations, though erratic absorption (oral suspension formulations) and nonlinear pharmacokinetics can make it difficult to assess the drug exposure of some azoles, increasing the need for therapeutic drug monitoring. Azole enzyme inhibition is not entirely limited to fungal cells and the class is associated with cytochrome P450 inhibition driving dangerous drug–drug interactions and “off-target” side effects in some cases [4,5,6,7]. The class is also restricted by a wide range of significant toxicities, including hallucinations, hepatotoxicity, and QTc prolongation [8]. Azole use has also been affected by recent increases in primary and acquired resistance among several fungal species.

In addition to ergosterol, another potent antifungal target is 1,3-β-d-glucan, an important component of fungal cell walls. Echinocandins inhibit 1,3-β-d-glucan synthesis to weaken fungal cell walls and trigger cell lysis (Figure 1). Echinocandin coverage is primarily limited to yeasts and moulds and they have little activity against endemic mycoses, and they remain one of the preferred treatment options for invasive candidiasis, including candidemia. Echinocandins are notably well-tolerated with limited adverse effects or drug interactions. Use is instead limited by a lack of oral formulations, with all current echinocandins only available as once-daily intravenous infusions.

Allylamines such as terbinafine interfere with ergosterol synthesis by the inhibition of squalene epoxidase. Commonly utilized as a topical agent, terbinafine is only used in the treatment of dermatophytes and moulds in the salvage setting, given the efficacy and more favorable adverse effect profile of other antifungal agents [9].

The final agent of clinical significance, flucytosine, is a pyrimidine analogue that selectively interferes with fungal nucleic acid synthesis to achieve activity [3]. It is available as an oral formulation and its major adverse effects are essentially limited to bone marrow suppression. However, the use as monotherapy is rare due to the rapid development of resistance, and it is primarily utilized as a component of combination therapy for the management of cryptococcal meningitis, urinary candidiasis, or chromoblastomycosis.

Accounting for spectra, toxicities, and formulations, the existing antifungal options still leave gaps in management and significant opportunities for the development of new agents. Some development has been focused on adding new agents within the existing classes (rezafungin) in addition to improving formulations of previously approved agents (SUBA-itraconazole, Amphotericin B cochleate). Novel classes of antifungals are also being actively developed. Ibrexafungerp and the tetrazoles act on similar fungal biosynthesis pathways previously targeted by existing classes (1,3-β-d-glucan and ergosterol synthesis), while many other agents in development (olorofim, MGCD290, Fosmanogepix, VL-2397, T-2307) look to establish entirely novel targets of antifungal activity.

## 2. SUBA-itraconazole

While significant effort has been made in identifying novel antifungal compounds and classes, there has also been progress in optimizing the agents within the present antifungal arsenal. Itraconazole is a broad spectrum triazole, but its clinical utilization has been hampered by severely limited bioavailability [10,11]. Traditional oral itraconazole capsules have a bioavailability of approximately 55% when administered with food, but this is reduced in patients with an elevated gastric pH (i.e., on acid suppression therapy) and exhibits significant variability between patients. This has driven the development of super-bioavailability-itraconazole (SUBA-itraconazole; Mayne Pharmaceuticals; Table 1). This oral capsule formulation utilizes the solid dispersion of itraconazole in a pH-dependent polymer matrix to enhance dissolution and absorption. It has been shown to significantly increase oral bioavailability (173%) while reducing interpatient variability compared to traditional oral formulation [11]. Additionally, there is little food or acid effect on the bioavailability with this novel formulation, a significant advance over conventional itraconazole [12].

The phase III study of SUBA-itraconazole compared it to conventional itraconazole therapy in the management of endemic mycosis (NCT03572049). A study evaluating its use as prophylaxis in patients undergoing stem cell transplantation versus a historical cohort of conventional itraconazole found that the SUBA-itraconazole formulation had significantly higher therapeutic levels, with improved gastrointestinal tolerability [13]. SUBA-itraconazole has subsequently been approved by the FDA for the indications of blastomycosis, histoplasmosis, and aspergillosis (in patients intolerant of or refractory to amphotericin B therapy). It is currently available as a 65 mg oral capsule, with a recommended maintenance dose of 130 mg once daily with food and a potential loading dose of 130 mg three times daily for the first 3 days.

## 3. Rezafungin

Rezafungin (CD101; Cidara Therapeutics; Table 1) is a next-generation echinocandin in development that shares its mechanism of action, the inhibition of 1,3-β-d-glucan synthesis, with other members of this class [2]. It is a structural analog of anidulafungin with modifications resulting in increased stability and solubility. It is extensively distributed in tissues and displays a significantly increased half-life (~133 h) in comparison to other echinocandins, allowing for significant pharmacokinetic advantages. Dose-proportional pharmacokinetics have been demonstrated with little interpatient variability and a favorable safety profile [14].

A recently completed two-part phase II study of rezafungin compared two loading doses of rezafungin to caspofungin in the treatment of candidemia/invasive candidiasis (STRIVE; NCT02734862) followed by possible oral fluconazole step-down [2,15]. Rezafungin 400 mg for the first week, followed by 200 mg once weekly, was identified as the most effective dosing scheme, and is the regimen currently under investigation in the ongoing phase III invasive candidiasis study (ReSTORE; NCT03667690).

Clinical trials to date have reported a desirable safety profile, in line with the rest of the echinocandin class [2]. It has been well-tolerated with minimal adverse events and no significant from standard of care comparator groups. 

The efficacy and spectrum of activity is again comparable to other echinocandins with potent coverage of many clinically relevant *Candida* spp., including *C. albicans*, *C. krusei*, and *C. tropicalis*, in addition to some *Aspergillus* spp. [2]. *FKS* mutations that impart echinocandin resistance has been reported to also impact rezafungin MICs, although not in all isolates. Broad cross-resistance was seen between rezafungin, caspofungin, and anidulafungin, although rezafungin’s “front-loaded” dosing regimen utilized in studies is suggested to lower development of resistance [16]. In vitro studies have also found rezafungin to have potent activity against *C. auris*, superior to both caspofungin and micafungin [17]. In vivo animal studies have also suggested that rezafungin, uniquely within the class, may be capable of preventing infections with *Pneumocystis* spp.

Given the broad activity of rezafungin, there is interest in its use for antifungal prophylaxis against *Candida*, *Aspergillus*, and *Pneumocystis*. As such, in a planned phase III trial, patients will be randomized to rezafungin or standard of care (triazole plus trimethoprim/sulfamethaxozole) (ReSPECT; NCT number pending) in patients undergoing blood or marrow transplantation. All studies will utilize rezafungin’s desirable once-weekly dosing regimen.

## 4. Ibrexafungerp

Ibrexafungerp (SCY-078; Scynexis, Inc.; Table 1) is another first-in-class agent (antifungal terpenoid) wielding a familiar mechanism of action [18]. Like echinocandins, it inhibits 1,3-β-d-glucan synthesis to achieve antifungal effect (Figure 1). However, as a triterpenoid enfumafungin derivative, ibrexafungerp is structurally unique from the echinocandin class [18,19].

As a compound, ibrexafungerp is highly bioavailable and can be administered as either an oral or intravenous formulation [18]. Taking the medication with food increases gastric dissolution and systemic absorption. Current phase III studies of oral ibrexafungerp utilize an initial loading dose of 750 mg PO BID for the first two days, followed by 750 mg PO daily for subsequent doses (NCT 03059992). Phase II studies previously utilized a 1250 mg PO once loading dose. It has been well-tolerated in studies and primarily limited to gastrointestinal adverse effects, with several studies identifying no significant difference from placebo arms [20].

The spectrum of activity, like echinocandins, includes a broad range of clinically significant *Candida* spp., including *C. glabrata* and *C. auris* [18]. Despite similar mechanisms of action, ibrexafungerp maintains in vitro activity against echinocandin-resistant *Candida* strains, suggesting a difference in target site avidity. Additionally, in vitro studies have identified fungistatic activity against *Aspergillus* spp., including azole-resistant strains [21]. However, ibrexafungerp, like echinocandins, lacks significant activity against agents of mucormycosis, although it displays some activity against historically challenging fungal species like *Lomentospora prolificans* and *Paecilomyces variotii*.

In phase II studies, oral ibrexafungerp was assessed as a therapeutic option in the management of moderate to severe vulvovaginal candidiasis as well as oral step-down therapy following initial echinocandin treatment [18]. It is currently being evaluated in an open-label, non-comparator, single arm phase III study as treatment of invasive fungal disease refractory or intolerant to standard-of-care therapy (FURI; NCT03059992), a phase III study vs placebo in patients with recurrent vulvovaginal candidiasis (CANDLE; NCT04029116). Interim analysis of the study has identified clinical benefit in 17 of the 20 evaluated patients [20].

Ibrexafungerp appears to have some functional similarities to the echinocandin class, including potent yeast activity and a desirable safety profile. However, as a new structurally distinct class, it also offers broader antifungal coverage, including moulds and echinocandin-resistant yeast, as well as a valuable oral formulation. Ongoing clinical studies will continue to define the clinical role of ibrexafungerp, but development appears to be targeted towards resistant fungal infections or oral step-down therapy following treatment with echinocandins.

## 5. Olorofim (F901318)

Another agent in development, olorofim (F901318; F2G Ltd.; Table 1), establishes a new antifungal class known as the orotomides with a novel mechanism of activity [2]. The class inhibits dihydroorotate dehydrogenase, a key enzyme in pyrimidine biosynthesis (Figure 1). The inhibition of pyrimidine production negatively affects fungal nucleic acid, cell wall, and phospholipid synthesis, as well as cell regulation and protein production. Olorofim displays time-dependent antifungal activity, with C_min_/MIC currently utilized as an indicator of pharmacodynamic efficacy.

Olorofim can be administered orally and intravenously, although the oral formulation has been the primary target of most studies. Current pharmacokinetic studies have found olorofim to be widely distributed throughout tissues, including even brain tissue, though at lower levels. It is cleared through the CYP450 (primarily CYP3A4) system, making it vulnerable to drug–drug interactions. It is not a CYP450 inducer itself but displays weak CYP3A4 inhibition. An oral dosing currently being studied is a loading dose of 4 mg/kg divided into two or three doses on Day 1, followed by a maintenance doses of 2.5 mg/kg/day divided into two to three doses per day as well with dose adjustments based on serum drug level. A C_min_ target of 0.5 mg/mL is based on in vivo pharmacodynamic studies against *Aspergillus* spp. [22]. Intravenous dosing was evaluated earlier in development to target the above serum target but has not been pursued significantly due to clinical inconvenience [2,23].

Olorofim possesses a broad spectrum of activity against moulds and appears to be particularly active against *Aspergillus* spp. [2,24]. Strong activity has been established against common *Aspergillus* spp. (*A. fumigatus*, *A. nidulans*, *A. terreus*, and *A. niger*) as well as cryptic species (*A. lentulus*, and *A. calidoustus*) that are historically difficult to treat and may be intrinsically resistant to many currently available antifungal classes. Olorofim was effective against multi-drug resistant *Aspergillus* strains, indicating a lack of cross resistance due to its novel mechanism of activity. Additionally, olorofim exposure did not appear to readily induce resistance in *A. fumigatus* samples [2]. It displays activity against uncommon moulds, including *Lomentospora prolificans* (for which there is currently no other effective therapeutic alternative) and *Scedosporium* spp. [2,24]. In vitro and in vivo activity against *Coccidioides* and other endemic mycoses has been identified as well [25]. Despite effective activity among the aforementioned fungal species, olorofim appears to possess minimal or no activity against *Candida* spp., Mucorales spp., and *Cryptococcus neoformans*, with variable and often species-specific results against *Fusarium* spp. [2].

There is an ongoing open-label phase IIb study evaluating olorofim in the treatment of susceptible invasive fungal infections among patients with limited treatment options (FORMULA; NCT03583164). This is in line with olorofim’s targeted role in practice as therapy for patients with invasive fungal infections lacking therapeutic alternatives, or inherently resistant or traditionally difficult to treat organism. Early success has seen olorofim granted breakthrough designation by the FDA and several phase III studies are in various stages of development.

## 6. MGCD290

MGCD290 (Mirati Therapeutics; Table 1) is an oral Hos2 fungal histone deacetylase (HDAC) inhibitor, that also affects non-histone proteins such as Hsp90 (Figure 1) [3]. HDACs and Hsp90 are a group of enzymes that play important roles in gene regulation and the control of cellular functions. MGCD290 appears to exhibit some level of intrinsic antifungal activity, but most research has pursued its value in synergizing with other antifungal agents. The inhibition of these fungal proteins could impair the cellular stress response, possibly potentiating the fungicidal effect of agents that target fungal cell wall or membrane. Several in vitro studies have found that the addition of low concentrations of MGCD290 enhanced both azole and echinocandin activity against strains of *Candida* spp. and *Aspergillus* spp., reducing MICs and driving categorical shifts from resistant to intermediate or susceptible in a large number of samples [26,27]. Despite promising in vitro results, MGCD290 has thus far failed to show efficacy in vivo [3]. A phase II study evaluating MGCD290 as adjunct therapy to fluconazole in severe vulvovaginal candidiasis found no benefit over fluconazole monotherapy, though it appeared to be well-tolerated [28]. Failure of efficacy to translate from in vitro to human in vivo studies is not uncommon and can result from a variety of complex factors. However, at the present time, no further studies have tackled the evaluation of in vivo efficacy to revive MGCD290 as a potential adjunctive antifungal therapy.

## 7. Amphotericin B Cochleate

Amphotericin B cochleate (CAMB/MAT2203; Matinas BioPharma; Table 1) is a new oral dosage formulation belonging to the polyene class. Current polyene formulations such as amphotericin B deoxycholate and the various lipid formulations of amphotericin are only approved for administration via intravenous injection [29]. Infusion-related reactions and dose-dependent renal toxicity limit the use of amphotericin deoxycholate [30]. Unlike the currently approved polyene formulations, amphotericin B cochleate is stable against degradation by the gastrointestinal (GI) tract. Cochleates are made up of phosphatidylserine with phospholipid-calcium precipitates. Their multilayered structure forms a solid, lipid bilayer, which is configured into a spiral, with no internal aqueous space [31]. Following oral administration, the cochleate is absorbed from the GI tract and enters circulation; once the calcium concentrations in the cochleates are decreased, the spiral formation opens and releases the encapsulated drug into the cell (Figure 1). Various formulations are delivered, utilizing the cochleate formulation such as proteins, peptides, and anticancer drugs such as raloxifene, fistein, and doxorubicin [32]. The limitations of cochleate formulations include stability loss at a temperatures greater than 4 °C, precipitation during storage, and being costly to manufacture [32].

Successful oral administration of amphotericin B cochleate was demonstrated in a murine mouse model by Santangelo et al. *C. albicans* was administered intravenously to mice; the non-cochleate group received intraperitoneal deoxycholate amphotericin or oral liposomal amphotericin, and experienced 100% mortality. The group that received amphotericin B cochleate orally experienced 100% survival at Day 16 post infection. The mg/kg dose of amphotericin B cochleate in this mouse model ranged from 0.5 mg/kg/day up to 20 mg/kg/day [31]. Amphotericin delivered via the oral route offers a practical manner in which to deliver a broad spectrum antifungal that has previously required intravenous infusion and been limited by infusion-related reactions. Safety and efficacy are currently under investigation in the treatment of vulvovaginal candidiasis (NCT02971007) and a study for the oral treatment of cryptococcal meningitis is planned.

## 8. Tetrazoles (VT-1129, VT-1161, and VT-1598)

Tetrazole antifungals are novel azole-like compounds with higher affinity for fungal cell CYP51 rather than human CYP. Currently available triazoles may have higher rates of adverse effects and drug–drug interactions due to their interaction with human CYP450 enzymes [33,34]. VT-1129 (Mycovia Pharmaceuticals; Table 1) is being developed as an oral antifungal. Wiederhold and colleagues evaluated VT-1129 in a mouse model and its activity against *Cryptococcus*. VT-1129 has a long half-life, approximately six days. A loading dose and maintenance dose was provided and compared to a control, and they found improved survival and decreased fungal burden in the treatment group [33]. VT-1598 demonstrates activity against yeasts, moulds, and endemic fungi. Again, Wiederhold et al. investigated VT-1598 and its activity against *Coccidioides immitis* and *Cocci posadasii* using a mouse murine model. They found the MICs to be 0.5 and 1 mcg/mL, respectively, compared to 16 mcg/mL for fluconazole. Increased survival was found in the VT-1598 group with a lower fungal burden. Higher fungal burden was present in the group receiving the lower dose of VT-1598, the control or fluconazole [35]. MICs were determined in an in vitro study of VT-1598, VT-1161 and the following triazoles: fluconazole, itraconazole, posaconazole, and voriconazole for *Candida*-susceptible and azole-resistant *Candida albicans* strains. VT-1598 and VT-1161 displayed potent activity against the majority of strains, many of which were known to be fluconazole resistant with an MIC ≥8 mcg/mL. The *C. albicans* strains used in this study carried multiple mechanisms of resistance. When comparing VT-1589 to VT-1161 in this study, VT-1598 had lower MIC_50_ and MIC_90_ values [36].

## 9. Fosmanogepix (APX001)

Fosmanogepix (APX001; Amplyx; Table 1) is a prodrug that is metabolized into its active form manogepix (MGX, formerly APX001A), also known as E1210 [37]. It targets the fungal-specific enzyme Gwt1, responsible for an early step in glycosylphosphatidylinositol (GPI)-anchor biosynthesis (Figure 1) [38]. This agent has demonstrated potency against multiple yeasts, moulds, and endemic mycoses [39]. Low MICs have been reported for *Candida* spp. (0.06 mcg/mL), *Cryptococcus neoformans* (0.5 mcg/ml), *Aspergillus* spp. (0.03 mcg/ml), *Scedosporium* spp. ranging from 0.015 to 0.06 mcg/mL, and *Fusarium* spp. 0.12 mcg/mL [1]. A mouse model was used by Alkhazraji and colleagues to assess the in vitro activity of MGX in two immunosuppressed murine models. Increased survival and decreased fungal burden were demonstrated by MGX [38]. An ongoing phase 2 trial evaluating safety and efficacy in candidemia is ongoing (SURGE; NCT03604705).

## 10. VL-2397

Another novel antifungal is VL-2397, also known as ASP2397, (Vical Inc.; Table 1) an intravenous agent isolated from *Acremonium* species. Structurally similar to fungal siderophores, it is taken up into the cell through the specific siderophore iron transporter 1 (Sit1) and disrupts intracellular processes through an unknown mechanism [2]. Sit1 is not found in mammalian cells, and VL-2397 is therefore thought to produce toxicity which is highly selective for fungal cells even at high concentration. In vitro and in vivo animal studies have found VL-2397 to be effective in the management of *Aspergillus* infections, primarily focusing on *A. fumigatus* [40]. A phase II trial was initiated evaluating VL-2397 in as first-line treatment for invasive aspergillosis in immunocompromised adults with acute leukemia or recipients of an allogeneic hematopoietic cell transplant but was recently discontinued by Vical Inc (NCT03327727).

## 11. T-2307

T-2307 (Toyama Chemical; Table 1) is an arylamidine compound with a novel mechanism of activity [3,41]. It is selectively transported into fungal cells through a polyamine transporter, where it inhibits mitochondrial function to generate a fungicidal effect. In vitro and in vivo animal studies have demonstrated a broad spectrum of activity against pathogenic fungi, including *Candida* spp., *Cryptococcus* spp., and *Aspergillus* spp. The identified minimum inhibitory concentrations have been extremely low and some animal models have suggested that T-2307 may be superior to azole and polyene therapy in the management of invasive fungal infections [3]. T-2307 has not yet progressed to clinical trials as ongoing animal model and in vitro studies focus on continuing to establish the compound’s characteristics. However, studies to evaluate clinical efficacy in the treatment of cryptococcal meningitis are planned.

## 12. Miscellaneous

A number of other antifungal agents are in various stages of development including nikkomycin (a chitinase inhibitor), and agents repurposed after in vitro antifungal activity was identified (sertraline, auranofin, tamoxifen) [42,43,44,45,46]. These latter agents have been subject to a variety of early clinical trials and in vitro/in vivo studies; however, a clear path forward for their development is yet to be presented.

## 13. Conclusions

Invasive fungal infections continue to increase coincident with improvements in oncologic, rheumatologic, and autoimmune therapy. The growing number of patients at risk for invasive mycoses has complicated management due to drug–drug interactions and intolerance of the currently available antifungal agents. The emergence of resistance, either innate or acquired, also continues to pose a major obstacle to improving patient outcomes and reducing morbidity and mortality. New agents and classes are a welcome addition to the antifungal armamentarium and results of ongoing clinical trials are eagerly awaited.

## Figures and Tables

**Figure 1 jof-06-00028-f001:**
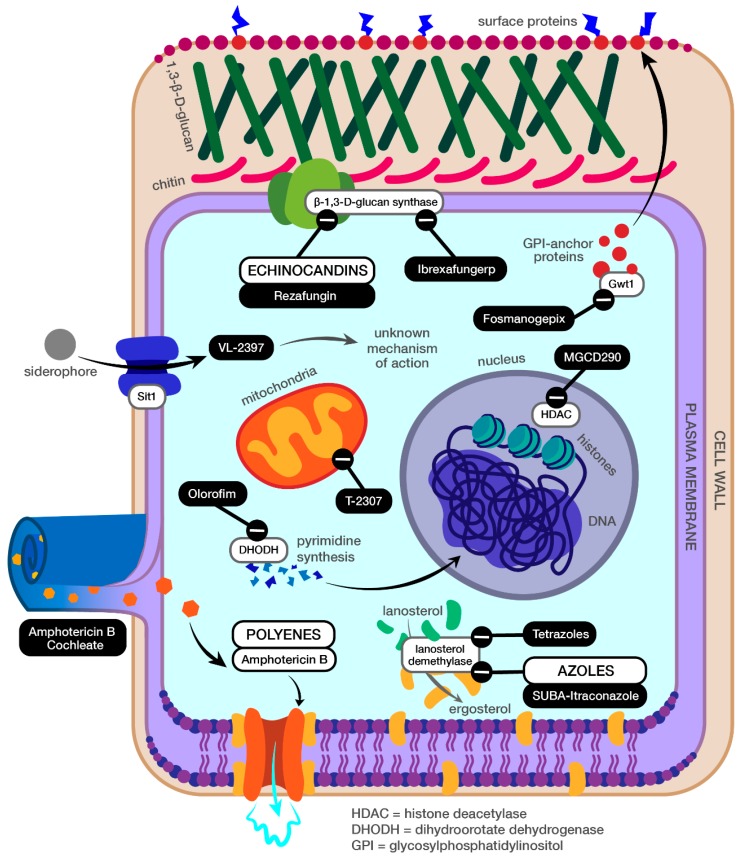
Developing antifungal agents and their mechanisms of action. Antifungal agents in clinical development utilize novel mechanisms of action or delivery systems to provide therapeutic alternatives to existing treatment options.

**Table 1 jof-06-00028-t001:** Summary of Aspiring Antifungal Agents and their Mechanism of Action, Spectrum of Activity, Phase of Development, and Clinical Advantages.

Class	Antifungal Agent	Mechanism of Action	Spectrum of Activity	Clinical Phase and Company	Clinical Advantages
Azole	SUBA-itraconazole	Interferes with cytochrome P450 activity, decreasing ergosterol synthesis, inhibiting cell membrane formation	blastomycosis, histoplasmosis, and aspergillosis	FDA approved Mayne Pharma Ltd.	Increased bioavailability compared to itraconazole
Echinocandin	Rezafungin	Inhibition of 1,3-β-D-glucan synthesis	*Candida albicans*, *Candida auris* *Candida krusei*, *Candida tropicalis*, *Aspergillus* spp. *Pneumocystis* spp.	Phase III Cidara Therapeutics, Inc.	Once-weekly dosing regimen. Treatment and potential role for prophylaxis
Terpenoid	Ibrexafungerp	Triterpenoid enfumafungin derivative that inhibits 1,3-β-D-glucan synthesis	*Candida* spp. including *Candida glabrata and Candida auris* *Aspergillus* spp.	Phase III SCYNEXIS Inc.	Oral and IV formulation Maintains activity against echinocandin-resistant *Candida* spp. and *Aspergillus* spp.
Orotomides	Olorofim	Inhibition of dihydroorotate dehydrogenase, thereby inhibiting pyrimidine production which negatively affects fungal nucleic acid, cell wall, and phospholipid synthesis, as well as cell regulation and protein production	*Aspergillus fumigatus, Aspergillus nidulans, Aspergillus terreus, and Aspergillus niger* and multidrug resistant strains of *Aspergillus* spp. Uncommon moulds such as *Lomentospora prolificans* and *Scedosporium* spp. Endemic Fungi	Phase II F2G Ltd.	Oral and IV formulation Activity against *Aspergillus* spp. including multidrug resistant and uncommon moulds
HDAC Inhibitor	MGCD290	Fungal histone deacetylase (HDAC) inhibitor	*Candida* spp. *Aspergillus* spp.	Phase II Mirati Therapeutics, Inc.	Possible role as an adjunctive antifungal in combination with an azole or echinocandin
Polyene	Amphotericin B Cochleate	Cochleate are a multilayered structure that forms a solid, lipid bilayer, configured into a spiral. Following oral administration, the cochleate is absorbed from the GI tract, enters circulation, and once calcium concentrations in cochleates are decreased, the spiral formation opens and releases the encapsulated drug into the cell.	*Candida* spp.	Phase II Matinas BioPharma	Oral formulation of amphotericin delivered via cochleate
Tetrazole	VT-1129	Interferes with cytochrome P450 activity, decreasing ergosterol synthesis, inhibiting cell membrane formation	*Cryptococus* spp. *Candida* spp.	Pre-clinical Viamet Pharmaceuticals, Inc.	May have reduced P450 drug interactions
	VT-1161	Interferes with cytochrome P450 activity, decreasing ergosterol synthesis, inhibiting cell membrane formation	*Candida* spp. Coccidioides spp. Rhizopus spp.	Phase III Mycovia Pharmaceuticals	May have reduced P450 drug interactions
	VT-1598	Interferes with cytochrome P450 activity, decreasing ergosterol synthesis, inhibiting cell membrane formation	*Candida* spp. including *C. auris* *Aspergillus* spp. *Cryptococcus* spp.	Phase I Mycovia Pharmaceuticals	May have reduced P450 drug interactions
Glycosylphosphatidylinositol inhibitor	Fosmanogepix (APX001)	Inhibits fungal Gwt1 GPI anchor protein. Low affinity for human GPI anchor proteins	*Candida* spp. including C. auris *Cryptococcus* *Coccidioides* *Aspergillus* and hyaline moulds Mucorales Not active against *C. krusei*	Phase II Amplyx Pharmaceuticals	Broad spectrum, oral formulation little toxicity in human studies thus far
Siderophore	VL-2397	Uptake via siderophore iron transporter	*Aspergillus* Some *Candida* spp. and *Aspergillus* spp. Mucorales	No current development plans – phase II trial terminated early	Activity against triazole resistant Aspergillus isolates
Arylamidine	T-2307	Thought to inhibit fungal mitochondrial synthesis	*Candida* spp. *Aspergillus* and some hyaline moulds	Phase I Toyama Chemical Company Ltd.	Structurally similar to pentamidine

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
