# Peer review of "Aspiring Antifungals: Review of Current Antifungal Pipeline Developments"

_jof, 2020, doi:10.3390/jof6010028_

Round 1
Reviewer 1 Report
An up-to-date review of the current antifungal drugs is discussed.
To increase the attractiveness of the review for the readers, I suggest adding a summarizing table in which some characteristics of the antifungals are compared, such as mode of action, activity spectrum, phase of development, limitations/side effects, …
Optionally, I would also suggest including a figure containing the known mechanisms of action of the discussed compounds.
Author Response
We agree and these are excellent suggestions. We have added a table and a figure summarizing novel antifungals currently in development.
Reviewer 2 Report
The authors have provided an extensive review on modified antifungal drugs based on old structures as well novel structures and their new targets. The article is clearly written and the subject is well studied. I can't find any faults apart from a few formatting errors.
Italic is required in line: 117 (in vitro), 172 (in vivo), 263 (in vitro), 291 (in vitro), 293 (in vivo/in vitro).
In vitro, in vivo or species names are not italicized in references.
Lines 208 and 243 could you check for additional spaces.
Line 246 "and" between species should be normal fond not italic.
For new antifungal compounds the company that developed them is mentioned except for VT compounds. Just for consistency you may want to mention Viamet.
Line 286/287: reads a bit odd (...to characterize the compound's characteristics...).
Author Response
We have made the suggested changes throughout the revised manuscript. We had initially not included Viamet as there were unsubstantiated rumors they had been acquired (we have not been able to confirm and thus left as Viamet in the manuscript).
Reviewer 3 Report
The review made by Gintjee et al. contains a well-written text with relevant information about the developments of new antifungals. To improve the quality of this review, some points should be revised:
-Lines 20-21: Clarify the following phrase: “This article reviews antifungal agents currently in development and undergoing clinical evaluation”. It is not clear if the purpose is to describe only the antifungals in clinical evaluation. Most antifungals included are in clinical phase, but T-2307 is only in pre-clinical studies;
-Lines 64-65: Indeed, the proposal needs to be clear. According to the introduction and conclusion, the antifungals discussed are focused on the invasive fungal infections, but the authors included topical antifungals (Allylamines);
-Line 72: Add a paragraph with the proposal of this review in the end of introduction section. Also, the antifungals that will be discussed need to be mentioned to give an orientation about how the subjects will be divided and described throughout the review.
-Line 216: Delete “The first polyene to market was amphotericin B deoxycholate in 1959”.
-Add a Table with the relevant information about each antifungal discussed, summarizing for example action mechanism, doses, and clinical phase.
Author Response
We have made all suggested edits/additions as requested and thank the reviewers for their time and attention to this paper. A revised manuscript with all 3 reviewers comments has been uploaded including a new table and figure.